# The New Potential of Deep Convective Clouds as a Calibration Target for a Geostationary UV/VIS Hyperspectral Spectrometer

**Yeeun Lee [1], Myoung-Hwan Ahn [1,*] and Mina Kang [2]**

[1] Department of Climate and Energy Systems Engineering, Ewha Womans University, 52 Ewhayeodae-gil, Seodaemun-gu, Seoul 03760, Korea; dungia@ewhain.net

[2] Department of Atmospheric Science and Engineering, Ewha Womans University, 52 Ewhayeodae-gil, Seodaemun-gu, Seoul 03760, Korea; mina@ewhain.net

[*] Correspondence: terryahn65@ewha.ac.kr; Tel.: +82-3277-4462

**Abstract:** As one of geostationary earth orbit constellation for environmental monitoring over the next decade, the Geostationary Environment Monitoring Spectrometer (GEMS) has been designed to observe the Asia-Pacific region to provide information on atmospheric chemicals, aerosols, and cloud properties. In order to continuously monitor sensor performance after its launch in early 2020, we suggest in this paper deep convective clouds (DCCs) as a possible target for the vicarious calibration of the GEMS, the first ultraviolet and visible hyperspectral sensor onboard a geostationary satellite. The Tropospheric Monitoring Instrument (TROPOMI) and the Ozone Monitoring Instrument (OMI) are used as a proxy for GEMS, and a conventional DCC-detection approach applying a thermal threshold test is used for DCC detection based on collocations with the Advanced Himawari Imager (AHI) onboard the Himawari-8 geostationary satellite. DCCs are frequently detected over the GEMS observation area at an average of over 200 pixels within a single observation scene. Considering the spatial resolution of the GEMS ($3.5 \times 8$ km$^2$), which is similar to the TROPOMI and its temporal resolution (eight times a day), the availability of DCCs is expected to be sufficient for the vicarious calibration of the GEMS. Inspection of the DCC reflectivity spectra estimated from OMI and TROPOMI data also shows promising results. The estimated DCC spectra are in good agreement within a known uncertainty range with comparable spectral features even with different observation geometries and sensor characteristics. When DCC detection is improved further by applying both visible and infrared tests, the variability of DCC reflectivity from TROPOMI data is reduced from 10% to 5%. This is mainly due to the efficient screening out of cold, thin cirrus clouds in the visible test and of bright, warm clouds in the infrared test. Precise DCC detection is also expected to contribute to the accurate characterization of cloud reflectivity, which will be investigated further in future research.

**Keywords:** GEMS; UV; VIS; hyperspectral data; deep convective cloud; vicarious calibration; OMI; TROPOMI

## 1. Introduction

With the global transport of anthropogenic chemicals in the atmosphere becoming a controversial issue over recent years, satellites have been considered a key tool for keeping track of chemicals given their wide spatial coverage. In the Asia-Pacific region, Geostationary Korea Multi-Purpose Satellite-2B (GEO-KOMPSAT-2B, GK2B) is expected to perform this role following its planned launch in February 2020 using an ultraviolet (UV) and visible (VIS) hyperspectral sensor called the Geostationary Environment Monitoring Spectrometer (GEMS). The GEMS has been designed to observe the Asia-Pacific region including the Korea Peninsula and surrounding areas and continuously monitor

atmospheric conditions by measuring the concentration of atmospheric chemicals and tracking aerosol properties [1,2]. To ensure the consistency of these measurements, the onboard calibration with solar diffusers and light-emitting diode (LED) is deployed in the GEMS calibration system, which converts light from a scene into calibrated spectral data (Level 1B). However, it has been frequently reported by previous satellite programs that residual errors in Level-1B data introduce some level of uncertainty to higher-level products [3–8]. It is also highly probable for a sensor's characteristics to change over time due to both internal and external factors, and this makes it necessary for the sensor to be continuously monitored and calibrated.

Vicarious calibration is a well-known approach for monitoring and improving sensor performance by periodically comparing it with reference targets. To successfully perform the calibration, it is important to select a suitable target that is stable enough to be repeatedly observed and well-characterized under different observation conditions. Because of these requirements, particular observation targets have been used for calibration, such as snow and ice over polar regions, bright clouds, deserts, and artificial sites [9–14]. However, geostationary earth orbit (GEO) sensors are limited in selecting the target because each sensor only covers a particular spatial region, while low-earth orbit (LEO) sensors cover the entire surface of the Earth. Considering that the GEMS only measures UV and VIS radiance reflected by the atmosphere and the Earth's surface, the variation in the measurements also imposes limitations on the selection of a stable target.

Deep convective clouds (DCCs), in this respect, are an excellent candidate as a calibration target for the GEMS considering their physical and radiative properties. DCCs are frequently observed over the tropical western Pacific (TWP) region with their tops reaching up to or over the tropopause due to the strong vertical convection [15–18]. This means that the backscattered radiation from these clouds is less affected by the Earth's surface and the troposphere, where most atmospheric components reside. The reflective properties of the cloud tops have also been fairly well-characterized due to their spatially uniform and less penetrative features, especially in the VIS and infrared (IR) spectral regions [19–22]. With these characteristics, DCCs have been widely used as a reference target for the monitoring of VIS and IR satellite sensors [23–31]. However, little attention has been paid to the applicability of DCCs as reference targets in the UV spectral region because there are not many UV sensors in operation, especially onboard GEO satellites. In this study, we aim to explore the applicability of DCCs as a reference target for the GEMS. Some of the advantages of using DCCs as a target are still valid even at shorter wavelengths, such as lower dependence on atmospheric conditions, the distinct brightness of the clouds, and the low spectral dependence in the reflected radiance from the clouds [32].

To select only spatially homogeneous clouds, we apply a DCC-detection routine with the IR brightness temperature (TB) threshold suggested by Doelling et al. [27] and an adaption of the UV–VIS threshold. Combining thermal and reflective signals is expected to facilitate the selection of suitable DCCs because each radiative property provides different types of information on the clouds [33,34]. In Section 2, to evaluate the applicability of DCC calibration, we firstly check whether DCCs occur over the TWP region in high enough numbers to provide reliable statistical parameters. Because the GEMS does not cover the IR region, we use TB and reflectivity data from the Advanced Himawari Imager (AHI) onboard a geostationary weather satellite (Himawari-8) to derive simple climatology for DCCs over the TWP region. After confirming that there are a sufficient number of DCCs over the TWP region, UV–VIS hyperspectral data from the Ozone Monitoring Instrument (OMI) onboard Aura and the Tropospheric Monitoring Instrument (TROPOMI) onboard Sentinel-5 Precursor (S5P) are used as a proxy for the GEMS for the spectral analysis of DCCs. In Section 3, we compare OMI and TROPOMI DCCs to confirm whether the detected DCCs reflect a sufficiently stable and bright signal to reduce the different sensor characteristics as having homogeneous spectral features. Based on these results, DCC detection thresholds are tested to optimize detection for further characterization of cloud reflectivity. In Section 4, we verify the effectiveness of the optimized DCC detection using TROPOMI observations and cloud properties from TROPOMI Level-2 data products. Preliminary results and the limitations of

our proposed method are also presented in this section. In Section 5, conclusions are presented with the remarks on the future research.

## 2. Data and Methods

### 2.1. UV–VIS Hyperspectral Sensor

#### 2.1.1. GEMS

The GEMS covers the Asia-Pacific region (5°S–45°N and 75°E–145°E), observing the Earth in an east-west direction with a fixed north–south field of view (FOV) of 7.73° [2]. For the retrieval of the concentrations of atmospheric gases ($O_3$, $NO_2$, $SO_2$, and HCHO) and aerosol properties, the GEMS has been designed to provide a continuous spectrum from 300 to 500 nm, with a spectral resolution of better than 0.6 nm every 0.2 nm. As the first hyperspectral UV–VIS sensor onboard a GEO satellite, the GEMS is expected to provide critical information for the monitoring of the regional transport of atmospheric chemicals at hourly intervals during the daytime as part of the GEO constellation [35].

Prior to the launch of the satellite, on-ground sensor characterization and calibration have been conducted during the preparatory phase for the GEMS. While in orbit, the GEMS relies on the onboard calibration consisting of solar diffusers and LED to evaluate and maintain calibration quality. As part of the onboard calibration system, the LED serves as a stable light source to monitor the non-linearity of the electronic response and the aliveness of each pixel at the detector level. Solar measurements have also been designed to monitor and calibrate changes in the sensor response with two transmissive diffusers: a working and reference diffuser. The working diffuser has been designed to observe the sun on a daily basis which makes it to gradually degrade over the course of the mission. Thus, a reference diffuser identical to the working diffuser but observing the sun only once every six months has been included in the calibration system. However, because most components of the sensor are expected to degrade over time, it is important to isolate the degradation of each component of the sensor and accurately calibrate the changes. Because onboard calibration has been incorporated into the calibration system, an independent method for evaluating the overall performance of the calibration system would be useful for maintaining the data quality of the GEMS in the long-term as a back-up calibration strategy.

#### 2.1.2. OMI and TROPOMI

The OMI and TROPOMI are hyperspectral sensors that encompass both the spectral range and the observation area of the GEMS. Operating in a sun-synchronous polar orbit, both sensors take radiance measurements in the ascending node of the satellites at around the local solar time (LST) of 13:30. The top-level specifications for the GEMS, OMI, and TROPOMI are summarized in Table 1. Launched in October 2017, the TROPOMI has stricter data quality requirements compared to other sensors. Because the spatial and spectral resolution of the GEMS is quite similar to the resolution of the TROPOMI, the GEMS and TROPOMI are strongly expected to be reciprocal candidates for inter-calibration once the GEMS goes into operation.

**Table 1.** Sensor specifications for the GEMS, OMI, and TROPOMI.

| Sensor | GEMS | OMI | | TROPOMI | |
|---|---|---|---|---|---|
| Orbit type | Geosynchronous (nadir at 128°E) | Sun-synchronous mean LST − 13:45) | | Sun-synchronous (mean LST − 13:35) | |
| Spectral range | 300–500 nm | UV-2 VIS | 307–383 nm 349–504 nm | Band 3 Band 4 | 320–405 nm 405–500 nm |
| Spectral resolution | < 0.60 nm | UV-2 VIS | 0.42 nm 0.63 nm | Band 3 Band 4 | 0.55 nm |
| Spectral sampling | < 0.20 nm/pixel | UV-2 VIS | 0.14 nm/pixel 0.21 nm/pixel | Band 3 Band 4 | 0.20 nm/pixel |
| Spatial resolution | $3.5 \times 8\ km^2$ (at Seoul) | $13 \times 24\ km^2$ (along × across track) | | $5.5 \times 3.5\ km^2$ (along × across track) | |

* The spatial resolution of TROPOMI Band 3-4 has been updated from 7 to 5.5 km along track since 6 August 2019 [36]. UV-2 and VIS indicate the Level 1B products of OMI while Band 3 and Band 4 indicate the products of TROPOMI.

### 2.2. DCC Climatology

To check whether there are sufficient DCCs available within the GEMS field of regards (FOR), especially over the TWP region, we apply a conventional DCC-detection approach using threshold tests for TB and the uniformity of the clouds [26]. The threshold values used for each test and the constraints for the observation angles and spatial coverage are summarized in Table 2. For the TB test, we use an 11-μm window channel with a threshold of 205 K, which is set considering the trade-off between the precision of DCC detection and the number of detected DCCs as presented by previous studies [27,28]. In addition, for the uniformity test, a relaxed threshold value (from 1 K to 2 K) is used to account for the lower spatial resolution of the GEMS. The relaxation of the threshold could broaden the range of available data with little change to the effectiveness of the DCC detection [28]. The maximum solar and viewing zenith angle is also limited to 40° because DCC reflectivity changes considerably when the solar and viewing angles are too large [21].

**Table 2.** DCC detection thresholds.

| Condition | Threshold |
|---|---|
| Infrared brightness temperature ($TB_{IR}$) | $TB_{IR} < 205$ K |
| Spatial uniformity ($TB_{IR}$) | Standard deviation of $TB_{IR} < 2$ K |
| Spatial uniformity ($R_{VIS}$) | Standard deviation of $R_{VIS} < 0.03$ |
| Solar and viewing zenith angle ($\theta_0$ and $\theta$) | $\theta_0 < 40°$, $\theta < 40°$ |
| GEMS observation area | 5°S–45°N, 75°E–145°E |

### 2.2.1. AHI Data Processing

AHI measurements are used because this imager onboard a GEO satellite provides VIS ($R_{0.47}$) and IR ($TB_{10.4}$) channels at a higher temporal resolution while fully covering the TWP region with its full-disk observation (see Table 3). Because $R_{0.47}$ has a higher spatial resolution than $TB_{10.4}$, spatially averaged $R_{0.47}$ is employed. To test the availability of DCCs under GEMS observation conditions, the spatial resolution of the GEMS is simulated using $4 \times 4$ pixels for each of the VIS and IR channels, while the mean of $TB_{10.4}$ and the standard deviations of $R_{0.47}$ and $TB_{10.4}$ are used for DCC detection.

**Table 3.** AHI VIS and IR channels for DCC detection.

| AHI | Ch01 ($R_{0.47}$) | Ch13 ($TB_{10.4}$) |
|---|---|---|
| Channel | VIS | IR (window channel) |
| Wavelength | 0.47 μm | 10.4 μm |
| Spatial resolution | $1 \times 1$ km$^2$ | $2 \times 2$ km$^2$ |
| Observation interval | Every 10 min | |
| Spatial coverage | Full-disk scan (nadir at 140.7°E) | |

### 2.2.2. Frequency Distribution

DCC climatology data from the AHI for July 2016 to June 2017 with a spatial grid and sampling frequency that matches that of the GEMS are collected. In the Asia-Pacific region, most DCCs are observed near the tropics and are distributed quite evenly over the GEMS observation area, as reported in previous studies [37–39]. In Figure 1, the spatial distribution of the frequency of DCCs exhibits a unique arc-shaped boundary, which is attributed to the limitations imposed by the current study (i.e., the viewing zenith angle (θ) should be smaller than 40°). Given that the viewing zenith angle is a fixed value over time for each pixel of a GEO sensor, the spatial distribution may be limited to those pixels that satisfy the angle condition.

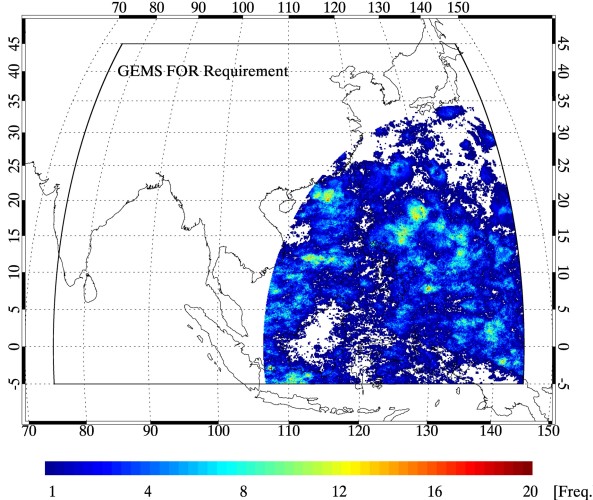

**Figure 1.** Frequency distribution of DCCs matched to the GEMS FOV over the GEMS observation area from AHI data taken at three-day intervals for the period July 2016–June 2017. Here, the frequency is calculated as the number of DCCs occurring over the year at three-day intervals at each AHI grid point binned to $8 \times 8$ km$^2$.

Figure 2 shows the temporal variation in the number of DCCs observed in a single observation scene. On a specific day, most DCCs are detected at noon (02:00–04:00 UTC) when the sun directly passes over the target area. This can be attributed to the constraint on the solar zenith angle for DCC detection because the solar zenith angle is also limited in the same way as the viewing zenith angle. The constraint along with the seasonal deep convection in the Northern Hemisphere might also cause the seasonal variation in the frequency of DCCs. As shown in Figure 2b, DCCs mostly occur from late summer to early autumn over the TWP region because atmospheric convection is strongly dependent on high moisture levels and the latent heat that accumulates during summer [38,40–42].

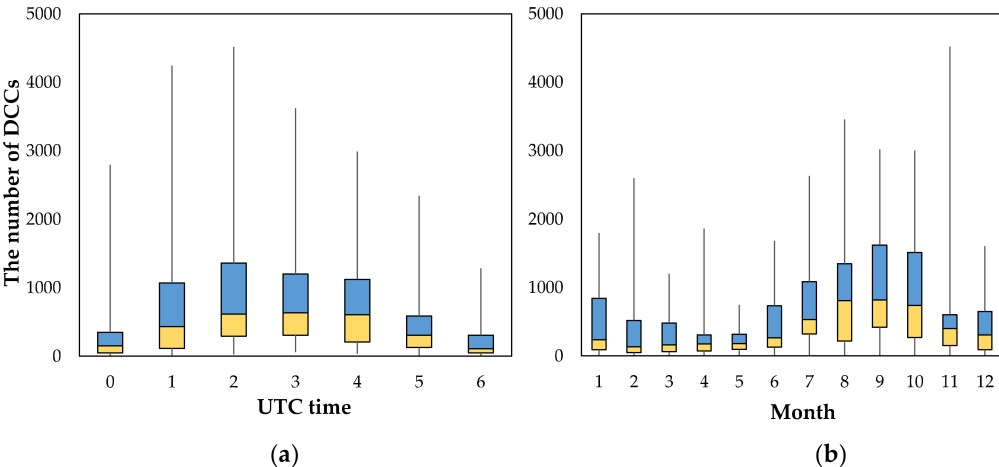

**Figure 2.** (**a**) Hourly and (**b**) monthly distribution of the number of DCCs observed in a single scene over the GEMS observation area corresponding to the GEMS FOV from AHI data taken at three-day intervals for the period July 2016–June 2017. The yellow and blue boxes represent the lower and upper quartile to the median, respectively.

Even with the limitation imposed by the viewing angular geometry and the seasonality, the average number of DCCs in a single observation scene is still larger than 200 pixels even in the month with the minimum frequency. Because the GEMS observes the Earth eight times a day, at least 50,000 DCCs can be detected a month from the GEMS when using the conventional DCC-detection approach with collocated AHI data. This number could be higher if the Advanced Meteorological Imager (AMI) onboard GEO-KOMPSAT-2A (GK2A) is used, which is stationed over 128.2°E as with the GEMS; thus, the coverage could be expanded further to the west.

*2.3. DCC Reflectivity Spectrum*

After confirming the availability of DCCs over the GEMS coverage area, DCC reflectivity spectra obtained from the OMI and TROPOMI are compared to confirm that the DCC measurements show similar spectral features and a sufficiently stable signal to be compared across different sensor characteristics and optical paths. In the UV–VIS spectral region, the reflected radiation from ice clouds is significantly affected by the angle condition [43], and this means it is important to precisely detect DCCs for the accurate characterization of cloud reflectivity.

2.3.1. Collocation Process

Because the OMI and TROPOMI only cover the UV–VIS and UV–SWIR spectral regions, respectively, DCC detection for both sensors could be performed with the collocated VIS and IR channels of the AHI. For the collocation between GEO and LEO sensors, we apply the collocation criteria suggested by the Global Space-based Inter-Calibration System (GSICS) community [44]. Because collocated LEO and GEO data are not directly compared in this study, the viewing angle does not match between the sensors. As shown in Figure 3, hyperspectral data satisfying the spatial and angle conditions (see Table 2) are collected first, and then the AHI VIS and IR channels matching the temporal collocation criteria are called. With the collected data, AHI pixels observed at nearly the same time ($\Delta t < 5$ min) as the OMI and TROPOMI pixels are collocated when the pixels simultaneously satisfy the spatial threshold (located within a half of shorter FOV of a LEO sensor). With the collocated AHI pixels, the average $TB_{10.4}$ and the standard deviations of $TB_{10.4}$ and $R_{0.47}$ are employed for DCC detection.

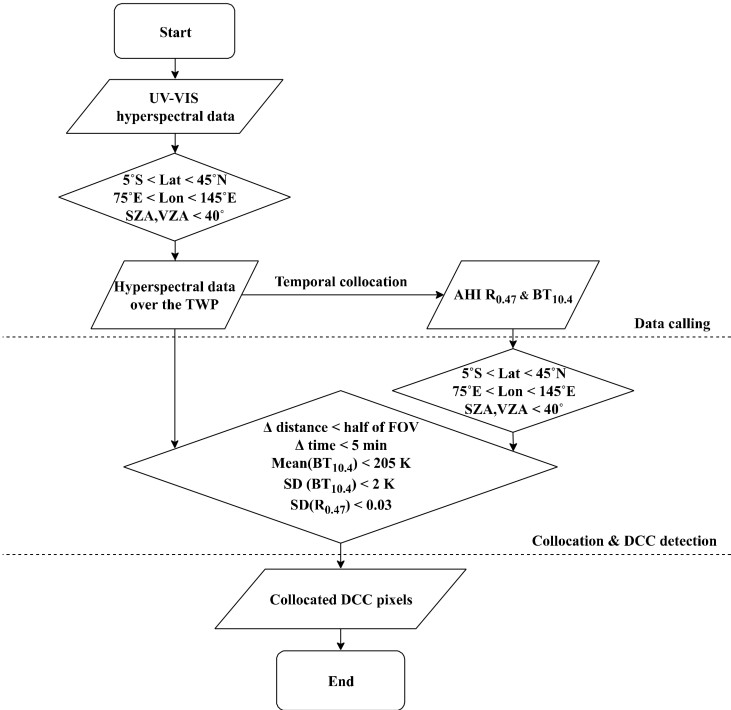

**Figure 3.** Flow chart of the collocation process between a UV–VIS hyperspectral sensor and a VIS–IR imager for the generation of DCC data. Mean and standard deviation (SD) are calculated with AHI pixels satisfying the spatiotemporal collocation criteria.

### 2.3.2. Apparent Reflectivity of DCCs

The GEMS, OMI, and TROPOMI provide the spectral radiance data that are used as input for the retrieval of geophysical information from the atmosphere. Because the uncertainty in the measured radiance due to the optical path of the instrument could be mitigated by using irradiance, which has the same optical depth [45], here, we use reflectivity for the spectral analysis. Because the OMI and TROPOMI provide solar observations once a day, timely matched irradiance with radiance is used to calculate the reflectivity. The radiance measured over the DCCs can be written as:

$$I_\lambda(\theta_0, \theta, \varphi) = R_\lambda(\theta_0, \theta, \varphi)\frac{F_\lambda}{\pi}e^{-(\frac{\mu+\mu_0}{\mu\mu_0})\tau_\lambda(z)} \tag{1}$$

where $I_\lambda(\theta_0, \theta, \varphi)$ is the measured upwelling radiance at wavelength $\lambda$ with solar zenith angle $\theta_0$, satellite zenith angle $\theta$, and relative azimuth angle $\varphi$. The measured radiance is strongly affected by cloud reflectivity $R_\lambda(\theta_0, \theta, \varphi)$ and incoming solar irradiance $F_\lambda$ at the top of the atmosphere (TOA). The equation also includes the attenuation caused by atmospheric extinction occurring when the incoming and outgoing radiation passes through the atmosphere. The atmospheric optical depth $\tau_\lambda(z)$ from the cloud top altitude $z$ to the TOA is determined by both absorption and scattering. Here, we consider only Rayleigh scattering to simplify the problem and neglect the backscattered radiation from the atmosphere above the DCCs. The angle component $\mu$ is the cosine of the zenith angle. Thus, cloud reflectivity using the measured radiance and irradiance can be given as:

$$R_\lambda(\theta_0, \theta, \varphi) = \frac{\pi I_\lambda(\theta_0, \theta, \varphi)}{F_\lambda}e^{(\frac{\mu+\mu_0}{\mu\mu_0})\tau_\lambda(z)} \tag{2}$$

Here, the optical depth $\tau_\lambda(z)$ is estimated using the approximation suggested by Bodhaine et al. [46] that considers the altitude and Rayleigh scattering in the atmosphere. The cloud altitude is set to approximately 16 km because the cloud top of DCCs nearly reaches the tropopause in the equatorial

region [31,47]. This means that the optical depth of the atmosphere above the clouds is within the range of 0.0005–0.0025 from 300 to 500 nm. Because Mie scattering and atmospheric absorption in the upper troposphere are not included in the calculation, the reflectivity is the apparent reflectivity of the DCCs, though it is referred to as simply DCC reflectivity in this study.

## 3. Results

### 3.1. DCCs Detected Using the OMI and TROPOMI

Figure 4 shows the DCCs identified using OMI and TROPOMI data for a particular cloudy scene (3 July 2018 06:10 UTC and 3 July 2018 06:40 UTC, respectively). The TROPOMI observes the Earth about 30 min earlier than the OMI, thus, the cloud distributions are slightly different. As shown in the figure, the number of DCCs obtained from the OMI–AHI collocations is appreciably smaller than that from the TROPOMI–AHI collocations. One of the main reasons for this difference is the lower spatial resolution of the OMI. An OMI pixel is about 15 times larger than a TROPOMI pixel, thus, many of the small-scale DCCs detected as DCCs using TROPOMI data are missed by the OMI because of the threshold and uniformity tests. The lower spatial resolution of the OMI for small-scale DCCs increases not only the TB but also the spatial variability in the IR and VIS channels, leading the pixel to be labeled as a non-DCC.

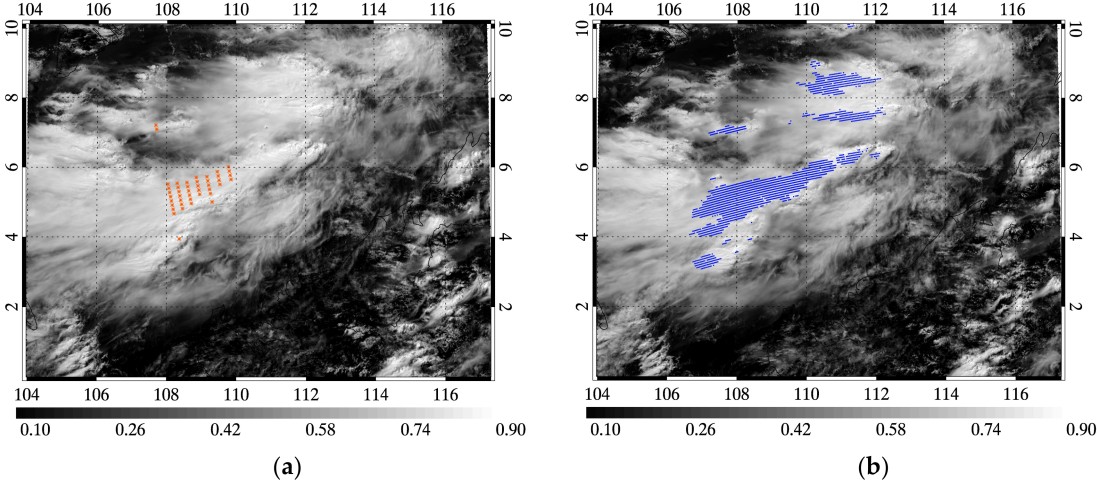

**Figure 4.** DCCs plotted on an AHI $R_{0.47}$ image: (**a**) OMI DCCs (orange dots) on 3 July 2018 06:10 UTC and (**b**) TROPOMI DCCs (blue dots) on 3 July 2018 06:40 UTC.

Data quality issues that arise during the long-term operation of the OMI also affect the availability of OMI observations. For instance, the row anomaly (RA) effect [32] renders nearly a quarter of all OMI pixels (especially those close to the nadir observations) unavailable for analysis. Figure 5 shows the measured reflectivity of the OMI as a function of the position (i.e., row) of the detector (i.e., the charge-coupled device, CCD) and the reflectivity spectrum affected by the RA effect. The measurements in rows 24–41 contaminated by the RA effect are eliminated during data processing. However, as shown in Figure 5a, the reflectivity for the row numbers close to the nadir port is also significantly lower, even though the rows are not flagged as RA-affected pixels. When these observations are detected together as DCCs, the reflectivity spectrum is significantly lower compared to the DCC reflectivity of the TROPOMI. Thus, in this study, the pixels in rows 41–48 are also eliminated, which are possibly affected by the RA effect but which are not flagged as such (https://projects.knmi.nl/omi/research/product/rowanomaly-messages.php). As shown in Figure 5b, with the elimination of the measurements in the CCD rows close to the nadir (41–48), the mean reflectivity becomes much closer to the DCC reflectivity of the TROPOMI. Because the rows close to

the nadir port generally have a low viewing zenith angle, which satisfies the angle condition for DCC detection, the RA effect significantly influences the availability of DCC observations from the OMI.

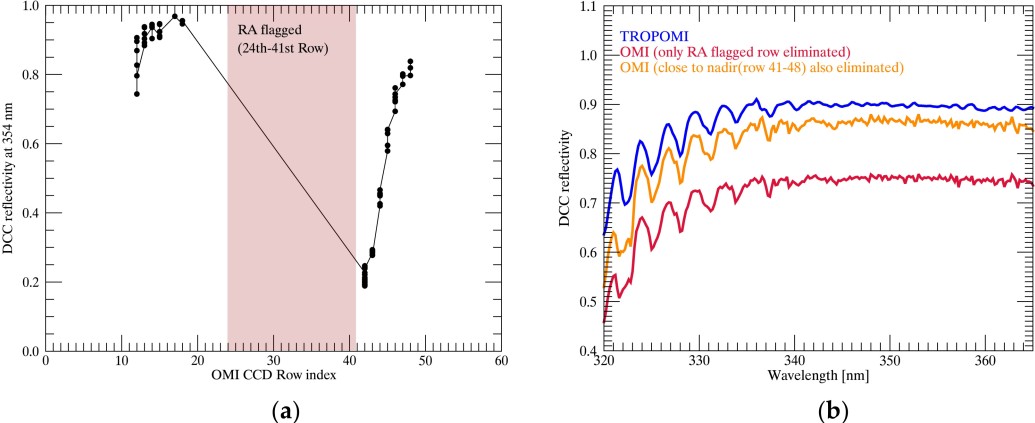

**Figure 5.** (**a**) DCC $R_{0.354}$ binned depending on the position of the detector in the OMI. The shaded red box indicates the RA flagged rows in the northern region of the orbit. (**b**) The DCC mean reflectivity spectrum of the OMI (the UV-2 product) containing the RA-affected observations in comparison with TROPOMI DCCs

### 3.2. DCC Reflectivity Spectrum

The DCC spectra of the OMI and TROPOMI observed over a year at 10-day intervals are presented in Figure 6, which shows the mean and standard deviation of the radiance and reflectivity at each wavelength. Solar measurements observed on the same day and the scan angle position (i.e., the position on the detector) of each DCC measurement are also displayed together. Because of the previously mentioned data quality issues, the number of DCCs from OMI observations over the year is only 3% of that from TROPOMI observations. However, even with this considerable difference in the number of measurements, the mean reflectivity of the OMI and TROPOMI is very similar at about 0.90 and 0.85, respectively, with nearly invariant spectral features except for both ends of the wavelength range. The spectral features at both ends are attributed to ozone absorption (300–345 nm) and the pixel saturation of the TROPOMI (450–500 nm) [36]. The results indicate that the DCCs observed by the satellite sensor reflect a mostly stable signal even with differences in sensor characteristics, the number of measurements, and the observation angle geometry.

However, some differences are also observed between the spectra of the OMI and TROPOMI. At shorter wavelengths, TROPOMI reflectivity is slightly higher than that of the OMI; as the wavelength increases, this difference becomes much smaller. This might be caused by the degradation of the diffuser in the solar measurements of the TROPOMI because this degradation is more significant at shorter wavelengths. This degradation is to be addressed in future updates of TROPOMI L1B data in early 2020, as announced in the S5P validation report [36]. TROPOMI DCCs also have less spectral noise because the OMI solar measurements have more spectral noise across all wavelengths, as shown in Figure 6a. There are also sharp peaks at 393 and 397 nm corresponding to the Ca II K and Ca II H Fraunhofer lines, respectively, which are caused by the beam-filling effect of the atmosphere above the clouds. Because rotational Raman scattering occurs in the atmosphere, scattered radiation is added to the upwelling radiation from the clouds [48]. However, OMI reflectivity exhibits negative peaks, which appears unrealistic considering that the beam-filling effect predominantly occurs with radiance. These peaks are caused by missing data at particular wavelengths for OMI irradiance. When calculating reflectivity, missing data are approximated by linear interpolation, which may not accurately reproduce the spectral features, especially for higher peaks.

These results indicate that the TROPOMI still requires further minor updates but that DCCs are a promising target given the theoretically well-matched spectral features and lower spectral

noise. The abundance of data is also an advantage of using the measurements in further research. However, even with the well-explained spectral features, DCC measurements still exhibit large systematic differences, as indicated by the standard deviations in Figure 6c, reaching nearly 10% and 12% for the OMI and TROPOMI, respectively. Because this systematic difference increases as the number of measurements increases, the TROPOMI has a higher systematic difference than the OMI. This indicates that, as the observation period becomes longer, the difference among the DCCs could increase considerably. The difference might be too large to regard the DCC detection properly done and this also make the characterization of the cloud reflectivity complicated without knowing the reason of the difference. Thus, in Section 3.3, the thresholds for conventional DCC detection are tested to reduce the systematic differences in DCC measurements and improve the accuracy of DCC detection. Because the OMI has some data-quality issues, we use only TROPOMI and AHI observations for this analysis.

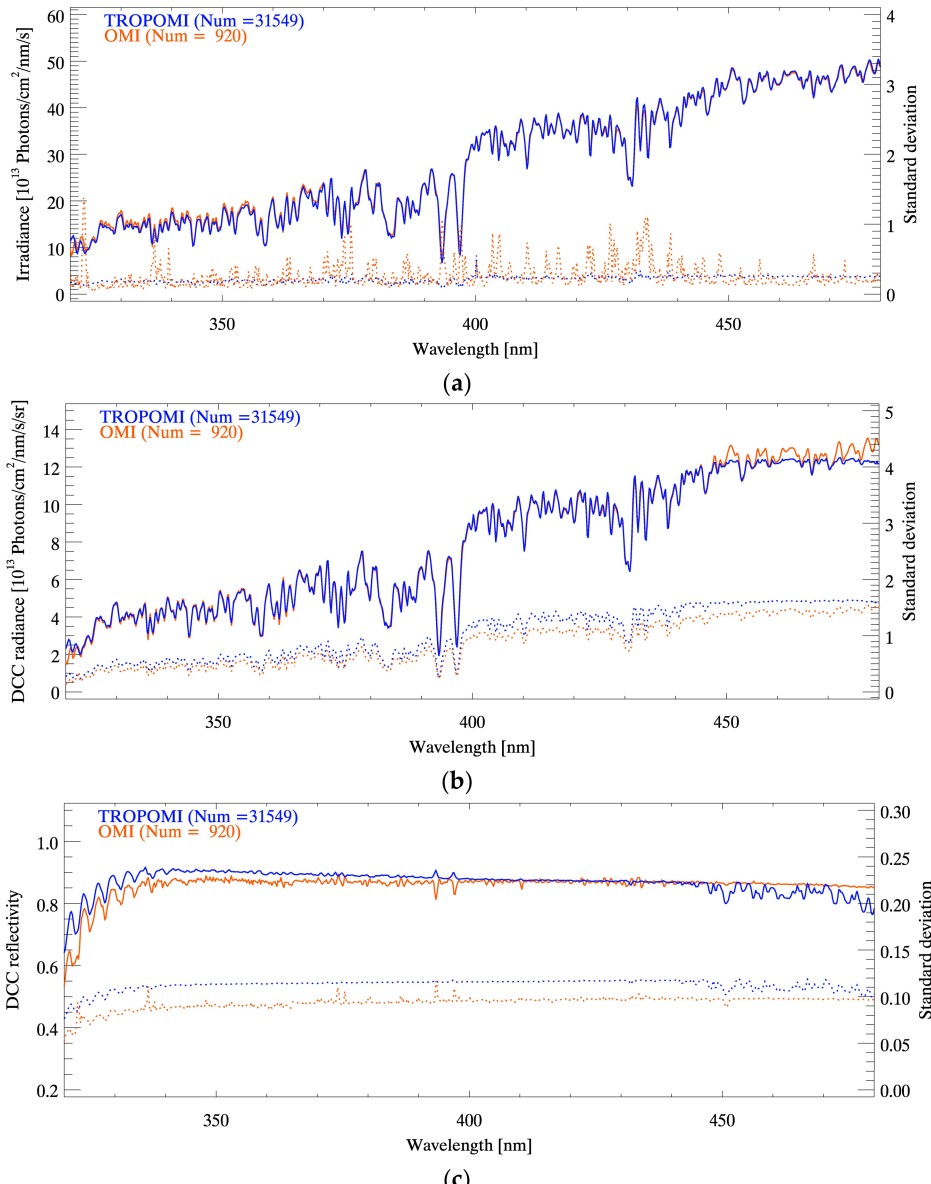

**Figure 6.** Mean and standard deviation of the (**a**) irradiance, (**b**) radiance, and (**c**) reflectivity spectra of OMI and TROPOMI DCCs observed for the period July 2018–June 2019 at 10-day intervals. The solid and dashed lines represent the mean and standard deviation at each wavelength, respectively.

### 3.3. Improvement in DCC Detection

#### 3.3.1. Comparison of VIS and IR Radiation

Figures 7 and 8 present the characteristics of the DCCs detected by the VIS and IR channels. This comparison provides insights into whether DCC detection is accurate when detecting only the colder and brighter cloud cores. Figure 7 shows the horizontal distribution of DCCs found over Typhoon Chaba in October 2016. For a one-to-one comparison, an AHI $R_{0.47}$ image is binned to match the spatial resolution of $TB_{10.4}$. As shown in Figure 7a, DCCs (identified as blue dots) are mainly found over the typhoon center, which has a cold $TB_{10.4}$, with a symmetrical distribution around the center. However, Figure 7b, which presents the DCCs detected over the $R_{0.47}$ image, is interesting in that the blue dots over the right side of the typhoon center have a lower $R_{0.47}$ of about 0.7. These are thin cirrus clouds that have spread out from the typhoon center following strong upper air outflow. Because these cirrus clouds have colder cloud tops composed of ice particles, the clouds are detected as DCCs using the conventional detection method even though their reflectivity is much lower than genuine DCCs.

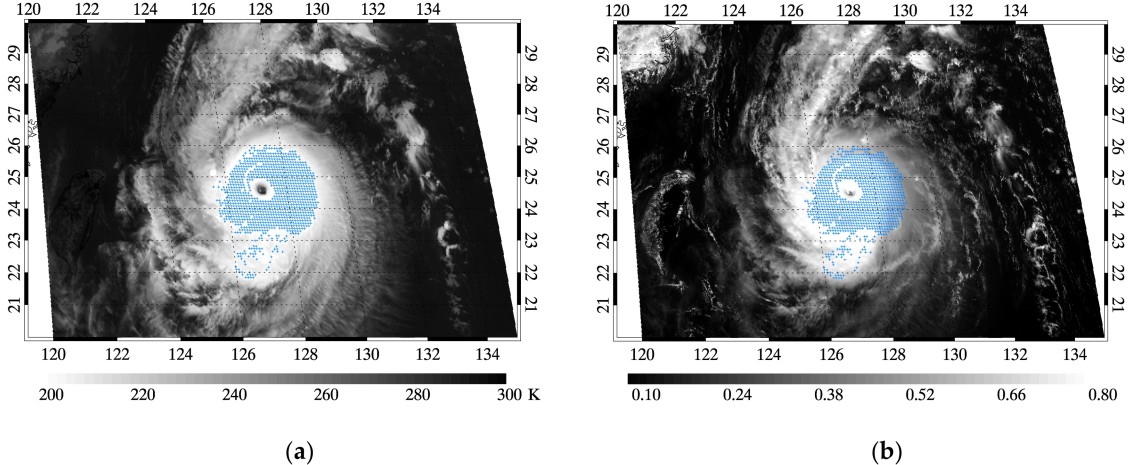

(**a**)                  (**b**)

**Figure 7.** AHI DCCs plotted as blue dots on (**a**) 2-km AHI $TB_{10.4}$ and (**b**) 2-km $R_{0.47}$ images of Typhoon Chaba (3 October 2016 03:30 UTC)

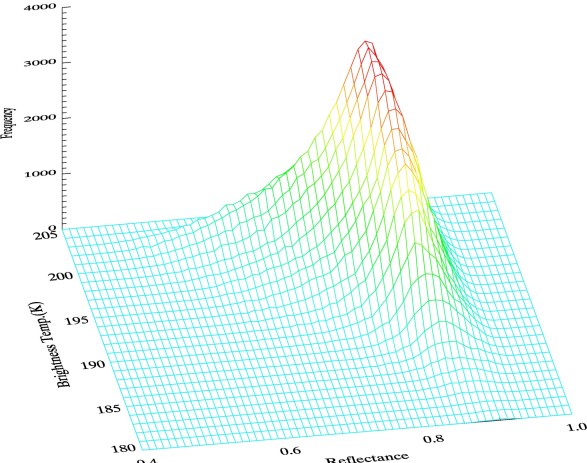

**Figure 8.** Two-dimensional histogram of DCCs detected using AHI $R_{0.47}$ and $TB_{10.4}$ over the GEMS observation area with AHI data taken at three-day intervals for the period July 2016–June 2017.

The difference in the radiative properties of the DCCs is also demonstrated in Figure 8, which presents a two-dimensional histogram of $R_{0.47}$ and $TB_{10.4}$ of the DCCs. Based on the histogram,

it can be inferred that an increase in $TB_{10.4}$ also increases the skewness of the distribution of $R_{0.47}$. This may be due to the increase in the proportion of detected DCCs with a lower reflectivity. This also indicates that DCCs usually have colder cloud tops, a higher reflectivity, and an optically thicker vertical structure, while cloud edges and cirrus clouds have similar colder but darker cloud tops. Consequently, these results show that $TB_{10.4}$ might be less effective as a DCC detection threshold, especially for UV–VIS measurements.

One of the few attempts to use DCCs for the monitoring of a UV–VIS hyperspectral sensor used only the UV reflectivity threshold for DCC detection [32]. In that study, OMI pixels with a higher reflectivity at 354 nm ($R_{0.354} > 0.9$) were identified as DCCs and then used for the monitoring of the temporal stability of the radiometric calibration of the OMI. At 354 nm, ozone absorption becomes weaker while Rayleigh scattering becomes stronger, and these interactions with the atmosphere reduce the proportion of the directly transmitted light from the clouds which shows higher angle dependence even though the dependence is not very significant over bright clouds [34]. Even with this simple form of detection, the average cloud reflectivity was fairly constant regardless of the wavelength (which is a characteristic of DCC reflectivity), and thus, they used DCC reflectivity for the long-term monitoring of spectral dependence in sensor performance. However, DCC reflectivity still exhibited seasonal and inter-annual variation, which was attributed to differences in cloudiness, angle dependence, and residual atmospheric effects (refer to Figure 32 in [32]). Although it is not easy to quantify, it is highly possible that these attributions could be increased when the detected DCCs are bright but low-lying warm clouds. For example, the optical path for warm clouds is much longer than that for DCCs, causing increased variability in the measured reflectivity due to the increased contribution from tropospheric air. By the same token, the angular variation of the measured reflectivity also increases with increasing optical depth.

To further clarify the issues associated with warm clouds, Figures 9 and 10 show the spatial distribution of $TB_{10.4}$ and $R_{0.47}$ in warm clouds and the spectral reflectivity of clouds with different $TB_{10.4}$ values, respectively. This demonstrates the importance of the IR threshold for the accurate detection of DCCs, especially in relation to high-altitude clouds with minimal influence from the troposphere. The blue dots in Figure 9 show DCCs with high reflectivity (TROPOMI $R_{0.354} > 0.9$) and warm IR temperatures (AHI $TB_{10.4} > 260$ K). In this case, most of the blue dots are located over the cloud edges with bright reflectivity, although their temperatures are much warmer than the nearby convection core. Thus, if we use the UV–VIS radiation threshold only, it would be difficult to screen out bright but warm clouds that are close to the cloud core.

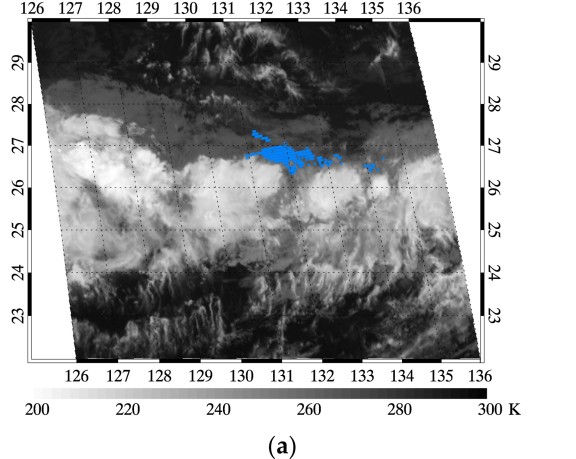
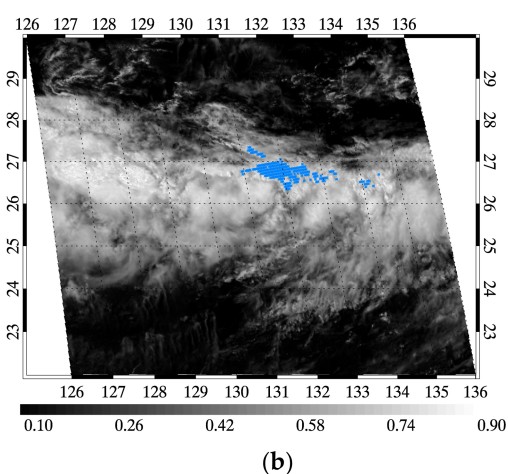

(a)  (b)

**Figure 9.** TROPOMI cloud pixels (AHI $TB_{10.4} > 260$ K, TROPOMI $R_{0.354} > 0.9$) plotted as blue dots on (**a**) 2-km AHI $TB_{10.4}$ and (**b**) 2-km $R_{0.47}$ images (20 June 2019 04:30 UTC).

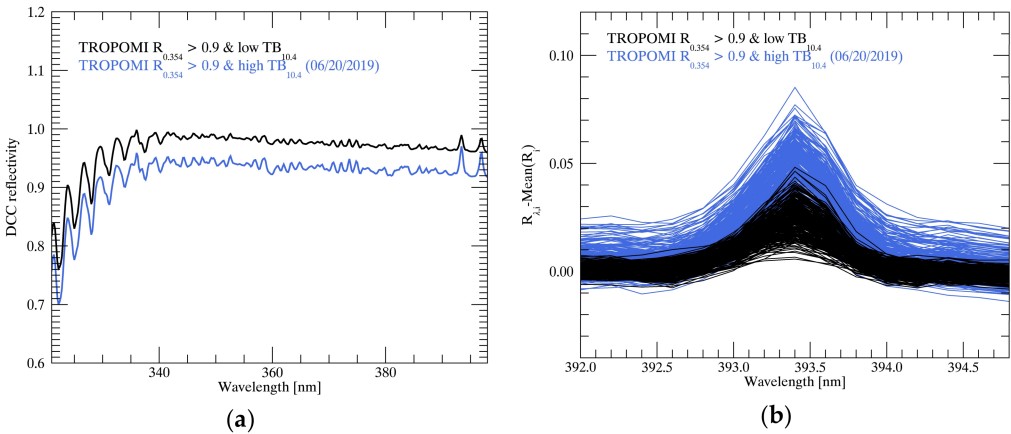

**Figure 10.** (**a**) Mean reflectivity spectrum of TROPOMI DCCs detected using UV reflectivity (TROPOMI $R_{0.354} > 0.9$) and (**b**) spectral anomaly spectra (i: each DCC pixel, λ: wavelength). The black and blue lines represent a cold and warm IR temperature, respectively.

Figure 10 shows the reflectivity spectrum of bright DCCs with different brightness temperatures. The blue line in Figure 10a represents the average reflectivity spectrum of the blue dots in Figure 9, while the black line represents that of the clouds satisfying the conventional DCC detection thresholds. The reflectivity spectrum including bright but warm clouds (blue line) clearly has a smaller reflectivity compared to the bright and cold clouds, which is attributed to the radiative interaction with the tropospheric atmosphere. The tropospheric effects in the measured reflectivity is also presented in Figure 10b because the beam-filling effect increases cloud reflectivity with greater rotational Raman scattering from the tropospheric atmosphere [48].

The results in Figure 8, Figure 9, Figure 10 make it clear that using VIS and IR information together could effectively screen out cirrus clouds and cloud edges as well as ensure the detection of only colder cloud tops for the better utilization of DCC reflectivity.

### 3.3.2. DCC Detection with Additional VIS Reflectivity

Based on the previous analysis, we develop an updated DCC detection approach utilizing both reflectivity and TB. In order to adapt the reflectivity test, it is important to set an appropriate threshold for reflectivity; a stricter threshold (e.g., 0.9) could produce more stable statistics but reduce the availability of the data, while a more relaxed threshold (e.g., 0.6) could increase the number of data points but increase the variability. Thus, the optimal reflectivity threshold for DCC detection needs to be set by weighing both sides (i.e., data availability and the stability of the reflectivity distribution). Here, we choose an optimal value by analyzing the variation in statistical parameters as a function of different threshold values.

Figure 11a,b presents the DCC frequency distribution for TROPOMI $R_{0.354}$ with the addition of the AHI $R_{0.47}$ threshold and the uniformity threshold for AHI $R_{0.47}$, respectively. The use of $R_{0.354}$ is based on a previous implementation with the OMI [32]. As shown in Figure 11a, applying the AHI $R_{0.47}$ test reduces the spread of the TROPOMI $R_{0.354}$ distribution and generates a distribution that closely follows a normal distribution. However, some low-reflectivity data remains because of the atmospheric effects and collocation uncertainty between the AHI and TROPOMI measurements. Figure 11b also shows that cloud pixels with higher spatial inhomogeneity account for a large proportion of the center of the distribution. This means that the overshooting tops near the cloud core may have a lower spatial uniformity, which cannot be eliminated by the reflectivity threshold.

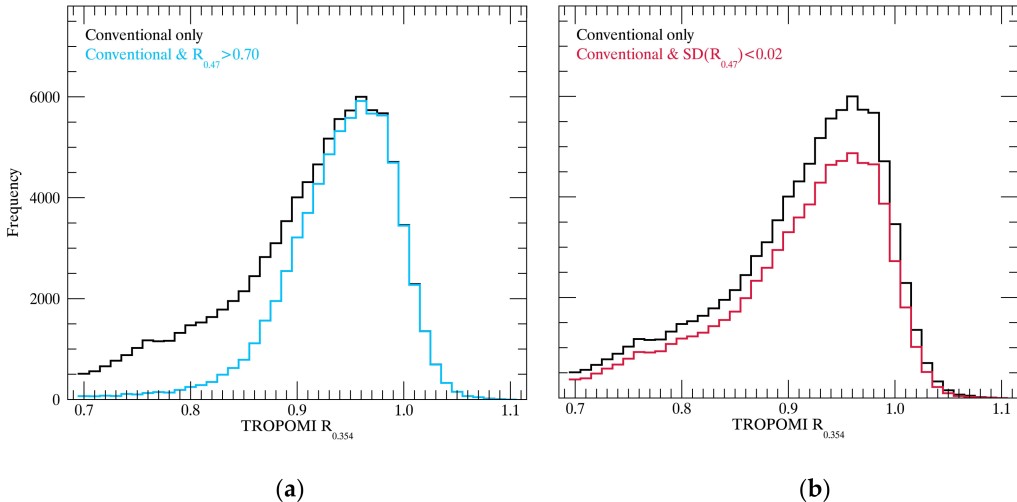

**Figure 11.** Frequency distribution of TROPOMI $R_{354}$ with an additional AHI $R_{0.47}$ restriction for data from July 2018–June 2019 taken at five-day intervals.

Table 4 presents the statistics for TROPOMI DCC $R_{0.354}$ with the application of different AHI $R_{0.47}$ thresholds to determine the optimal threshold that produces a fairly normal distribution without eliminating too many observations. TROPOMI $R_{0.354}$ is also applied together as the detection threshold to reduce collocation uncertainty by restricting the tail of the distribution (TROPOMI $R_{0.354} > 0.7$). The results show that, as the AHI $R_{0.47}$ threshold increases, the distribution becomes very close to normal even though the number of detected DCCs decreases exponentially. The standard deviation of the reflectivity decreases linearly and the VIS threshold increases when the kurtosis increases exponentially. Interestingly, only skewness converges at a particular AHI $R_{0.47}$ threshold (0.64). Because TROPOMI $R_{0.354}$ reflectivity is skewed to the left due to the darker cirrus clouds with a lower reflectivity, the skewness of the distribution has a negative value regardless of the AHI $R_{0.47}$ threshold.

**Table 4.** Statistics for TROPOMI DCC $R_{0.354}$ depending on the addition of an AHI $R_{0.47}$ threshold for DCC detection compared to the conventional DCC detection (w/o column) using DCC measurements for July 2018–June 2019 taken at five-day intervals.

| AHI $R_{0.47}$ | w/o | 0.60 | 0.62 | 0.64 | 0.66 | 0.68 | 0.70 | 0.72 | 0.74 | 0.76 |
|---|---|---|---|---|---|---|---|---|---|---|
| Count | 91630 | 90752 | 89861 | 88286 | 86138 | 83569 | 80475 | 76696 | 71469 | 64857 |
| Mean | 0.916 | 0.917 | 0.919 | 0.922 | 0.925 | 0.929 | 0.933 | 0.938 | 0.943 | 0.949 |
| Median | 0.932 | 0.933 | 0.934 | 0.936 | 0.938 | 0.940 | 0.943 | 0.946 | 0.951 | 0.956 |
| Mode* | 0.960 | 0.960 | 0.960 | 0.960 | 0.960 | 0.960 | 0.960 | 0.960 | 0.960 | 0.960 |
| SD* | 0.076 | 0.074 | 0.072 | 0.070 | 0.067 | 0.063 | 0.060 | 0.057 | 0.053 | 0.050 |
| Skewness | −0.779 | −0.769 | −0.765 | −0.761 | −0.767 | −0.780 | −0.803 | −0.848 | −0.917 | −1.021 |
| Kurtosis | -0.027 | -0.008 | 0.029 | 0.110 | 0.255 | 0.448 | 0.706 | 1.056 | 1.570 | 2.225 |

\* The bin size used to calculate the mode is set to 0.01. SD indicates the standard deviation.

Table 5 presents the statistics for TROPOMI DCC $R_{0.354}$ with the application of different thresholds for the uniformity test for AHI $R_{0.47}$. As shown in Table 5, the central value and the spread of the distribution changes only slightly with the different thresholds for the uniformity test for $R_{0.47}$. The kurtosis and skewness also change as the uniformity increases, though they do not change dramatically, as with the reflectivity threshold.

**Table 5.** Statistics for TROPOMI DCC $R_{0.354}$ depending on the uniformity threshold for AHI $R_{0.47}$ for DCC detection compared to the conventional DCC detection (w/o column) using DCC measurements for July 2018–June 2019 taken at five-day intervals.

| SD* of AHI $R_{0.47}$ | w/o | 0.025 | 0.024 | 0.023 | 0.022 | 0.021 | 0.020 | 0.019 | 0.018 | 0.017 |
|---|---|---|---|---|---|---|---|---|---|---|
| Count | 91630 | 84159 | 82454 | 80629 | 78626 | 76557 | 74241 | 71737 | 69135 | 66243 |
| Mean | 0.916 | 0.916 | 0.916 | 0.916 | 0.916 | 0.916 | 0.916 | 0.916 | 0.916 | 0.916 |
| Median | 0.932 | 0.933 | 0.933 | 0.933 | 0.933 | 0.933 | 0.933 | 0.932 | 0.932 | 0.932 |
| Mode* | 0.960 | 0.960 | 0.960 | 0.960 | 0.960 | 0.960 | 0.960 | 0.960 | 0.960 | 0.960 |
| SD* | 0.076 | 0.075 | 0.075 | 0.075 | 0.074 | 0.074 | 0.074 | 0.074 | 0.074 | 0.074 |
| Skewness | −0.779 | −0.788 | −0.789 | −0.792 | −0.793 | −0.795 | −0.797 | −0.796 | −0.795 | −0.797 |
| Kurtosis | −0.027 | −0.002 | 0.006 | 0.012 | 0.017 | 0.019 | 0.026 | 0.025 | 0.023 | 0.026 |

* The bin size used to calculate the mode is set to 0.01. SD indicates the standard deviation.

In summary, the skewness of distribution of TROPOMI $R_{0.354}$ might become close to 0 with a brighter AHI $R_{0.47}$ threshold until the number of DCCs is significantly lower. When it comes to spatial inhomogeneity, some DCCs with a relatively low uniformity are eliminated with the stricter uniformity test mostly at the center of the distribution. Because the AHI $R_{0.47}$ threshold and the uniformity test might simultaneously affect the statistics for the reflectivity distribution of TROPOMI $R_{0.354}$, the optimal threshold value for DCC detection needs to be set considering both effects. Figure 12a,b show the number of available DCCs and the skewness of the distribution, respectively, as a function of the detection thresholds. Considering the distribution of each variable, the optimal thresholds for AHI $R_{0.47}$ and the uniformity test for DCC detection are set at 0.70 and 0.018, respectively, because at that point, the available number of DCCs is still high even with a relatively low skewness of −0.70.

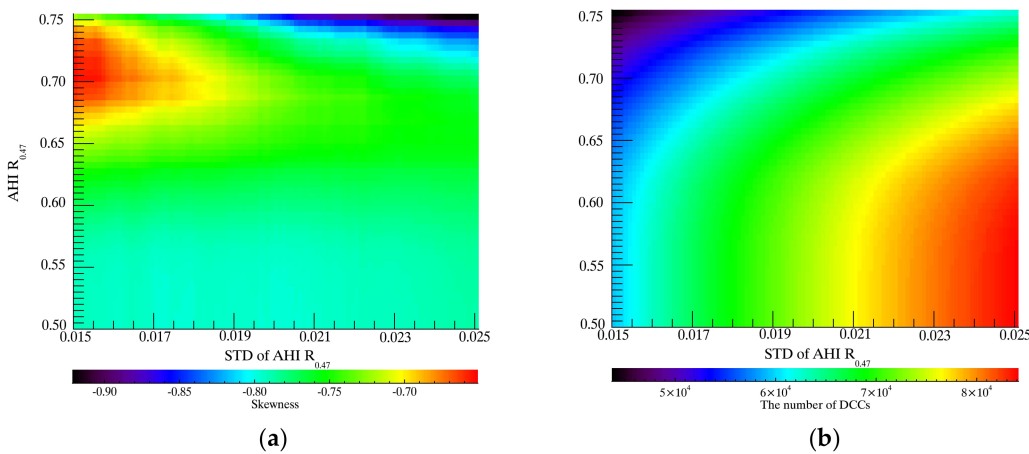

**Figure 12.** (**a**) Skewness and (**b**) the number of DCCs as a function of the AHI $R_{0.47}$ and uniformity test for DCC detection using DCC measurements for July 2018–June 2019 taken at 5-day intervals.

## 4. Discussion

### 4.1. Verification of the Updated DCC Detection Method

The results in Section 3 show that DCCs have different radiative properties depending on the way to detect the DCCs. For the thermal threshold test, it would be most effective to screen out the low-altitude clouds, in this case warm clouds having longer optical path lengths. VIS reflectivity can also be a useful indicator for detecting only optically thick clouds that are bright enough to reflect most of the incoming radiation. Using both radiative properties, DCC detection can be improved further to

detect only optically thick and high-altitude cloud targets that exhibit homogeneous spectral features and higher reflectivity with lower variation.

### 4.1.1. Spectral Analysis of DCC Reflectivity

Figure 13 highlights the advantages of applying the updated DCC detection method with the threshold values suggested in Section 3.3.2. The DCC mean reflectivity spectra at the Fraunhofer lines are presented to compare the spectral features of the DCCs detected by different detection methods, including the UV threshold test (TROPOMI $R_{0.354} > 0.9$) and the IR threshold test (AHI $BT_{10.4} < 205$ K). In Figure 13a, the mean reflectivity spectra show similar spectral features but differences in reflectivity as the DCCs detected using the UV threshold test show the highest values. However, in Figure 13b, the spectral features of anomaly spectra exhibit more variance when only the UV detection threshold is used. In the figure, the DCCs detected using the updated DCC detection method have lower peaks at the Fraunhofer lines, which indicates that the atmosphere above the clouds might be much thinner when the DCCs are detected using the thermal radiation threshold.

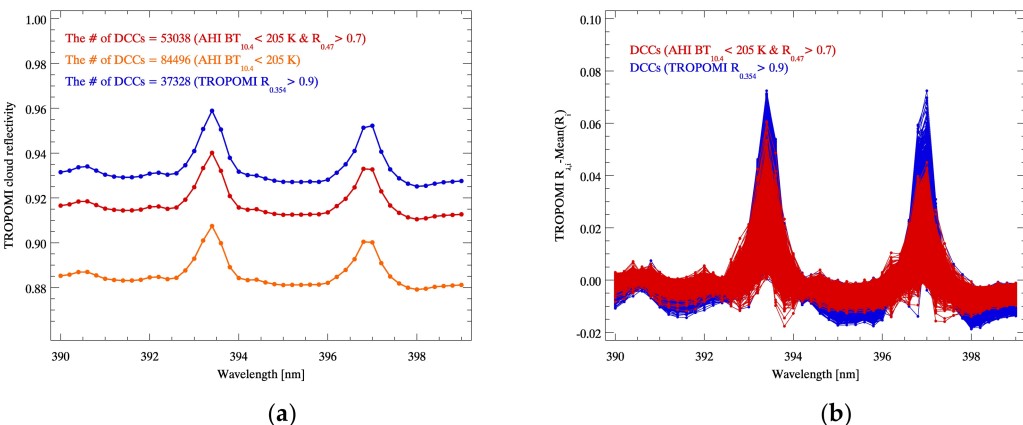

**Figure 13.** (**a**) Mean reflectivity and (**b**) anomaly spectra (i: each DCC pixel; λ: wavelength) of DCCs detected using different DCC detection threshold tests. The blue, red, orange lines represent the UV threshold test only, the updated DCC detection method, and the IR threshold test only, respectively. DCC measurements are from July 2018–June 2019 taken at five-day intervals.

### 4.1.2. Cloud Properties of DCCs

The cloud properties obtained from TROPOMI Level 2 cloud products are presented in this section in order to identify the practical range of cloud properties for the DCCs detected using different DCC detection threshold tests. Cloud optical thickness and cloud top height are used for this analysis because these properties represent the optical and physical features of the clouds, respectively. The cloud properties are retrieved from the $O_2$ A-band at 760 nm, while the clouds are treated as scattering layers [49]. In Figure 14a, the optical thickness of the DCCs detected using the IR detection threshold is lower than that of the DCCs detected using the UV detection threshold. However, as shown in Figure 14b, the cloud top height is much higher when the IR threshold test is used for DCC detection. These results indicate that the UV and IR DCC detection thresholds complement each other in limiting various cloud properties while accurately detecting only those DCCs with homogeneous cloud properties. These results closely correspond with the analysis in Section 3.3.

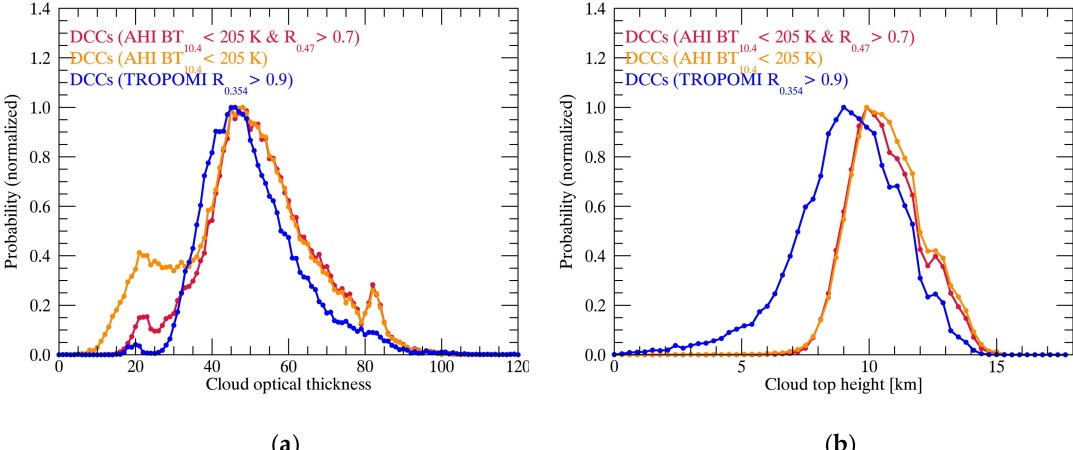

(**a**)　　　　　　　　　　　　　　　　　(**b**)

**Figure 14.** Same as Figure 13 for histograms of (**a**) cloud optical thickness and (**b**) cloud top height from the TROPOMI Level -2 cloud product for DCCs.

### 4.2. Feasibility and Limitations

In this section, we present the feasibility of using DCC calibration for a UV and VIS hyperspectral sensor based on our updated DCC detection method. As mentioned in Section 1, DCC calibration has been generally used with meteorological sensors to update radiometric calibration coefficients, which typically change over the course of the operation period. A meteorological sensor can be calibrated with the well-calibrated sensor after the normalization of various observation conditions, such as the angle dependence, spectral response functions, and different center wavelengths. DCC calibration for environmental sensors still has a long way to go in terms of normalization, but in this study, we present preliminary results for the temporal variability in the TROPOMI DCC observations.

Figure 15a presents the seasonal distribution of TROPOMI DCCs for data collected over the period of a year with probability density functions (PDFs). Even though the number of DCCs is not sufficient to calculate a representative PDF for the observations, the PDFs have similar distribution patterns regardless of the number of DCCs in each season. However, given that distribution modes are generally used to monitor the calibration accuracy of meteorological sensors, the PDF modes are too variable since the bidirectional reflectivity of the DCCs and the disparity in the cloud optical properties have not been sufficiently accounted for so far. However, the temporal variability caused by these uncertainties could cancel each other out as the reflectivity ratio between two different wavelengths represents in Figure 15b. The ratio of DCC reflectivity at 354 and 397 nm is used because reflectivity at 397 nm (Ca II H line) is affected both by scattered and directly transmitted light. Even with the highly expected variability, the ratio of the mean reflectivity at both wavelengths appears relatively stable within 0.99–1.01.

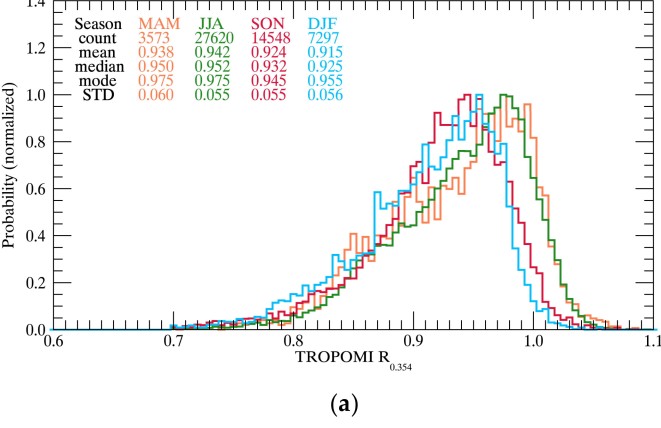

(**a**)

**Figure 15.** *Cont.*

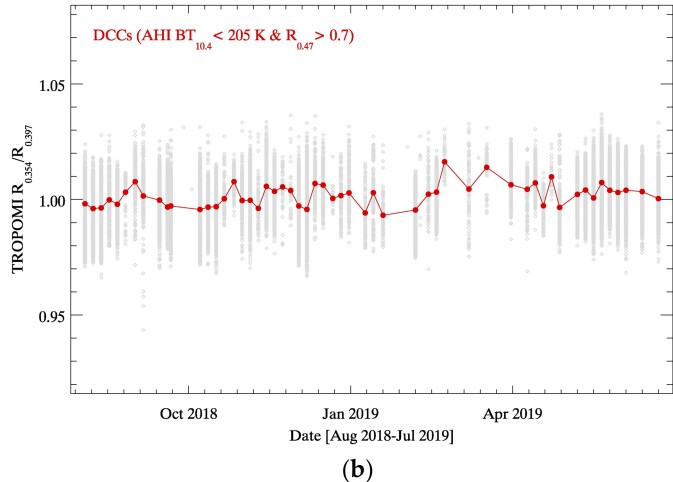

**Figure 15.** (**a**) Probability density function for TROPOMI $R_{0.354}$ over time (MAM: March to May; JJA: June to August; SON: September to November; DJF: December to February) and (**b**) time series of mean reflectivity ratio of $R_{0.354}$ and $R_{0.397}$ (grey diamonds are individual values) for the DCCs detected using the updated DCC detection method.

## 5. Conclusions

As the first UV–VIS hyperspectral sensor onboard a GEO satellite, the GEMS covers the Asia-Pacific region, including the TWP region. To develop a vicarious calibration approach based on the current availability of calibration targets, the present study tests DCCs to determine whether optically thick clouds provide a sufficiently stable and bright signal to allow the radiometric calibration of sensors with different hardware characteristics and observation conditions especially in the UV–VIS spectral region. For feasibility testing, the VIS and IR channels of the AHI are used with UV–VIS hyperspectral data from the OMI and TROPOMI, as a surrogate for the GEMS. To mitigate the calibration uncertainty caused by degradation and high-frequency perturbations of the instrument optical paths, reflectivity (i.e., the ratio between radiance and irradiance) is used. The cloud reflectivity is calculated by taking account of the solar zenith angle, the satellite zenith angle, and Rayleigh scattering above the clouds.

To ensure a sufficient number of DCCs over the GEMS observation area, AHI data from a year-long period that match the spatial and temporal resolution of the GEMS are analyzed. The DCCs detected using the conventional approach (i.e., thermal temperature tests and uniformity tests) have a clear seasonality, with a maximum in September and a minimum in April. Spatially, the viewing zenith angle also limits the number of DCCs because the AHI observes the target area with a higher viewing zenith angle compared to the GEMS. This limitation of the satellite zenith angle is expected to be improved with the AMI onboard GK2A, which has the potential to be collocated with the GEMS as stationed nearby at 128.2°E. Even with these limitations, DCCs occur in more than 200 pixels on average in a single observation scene, which appears to be sufficient for the proposed statistical approach considering the observation frequency and the spatial resolution of the GEMS.

Although the number of DCCs detected by the OMI and TROPOMI is significantly different, mainly due to the poor spatial resolution and degraded quality of OMI data, a comparison between the estimated spectral reflectivity of the DCCs shows comparable results even with clear differences in sensor characteristics, viewing geometry, and the number of data points. Given that more accurate calibration is essential for achieving the final goal of the mission, the results look promising in terms of applying the proposed method to various UV and VIS environmental sensors for inter-calibration. However, a closer inspection of the reflectivity spectra shows that there is high variability in the standard deviation (up to 10%), which is mainly due to the false classification of thin cirrus clouds as DCCs, which have a cold cloud temperature with a low optical depth. Furthermore, inspection of an alternative approach using only reflectivity tests for DCC detection leads to the false detection of

warm clouds having a high reflectivity and a lower cloud top altitude. Thus, we devise an updated DCC detection approach using both thermal and reflectivity tests to screen out cold, thin cirrus clouds and bright, warm clouds. Based on the variation in the statistical parameters of DCC reflectivity with different reflectivity threshold values, the threshold value for the reflectivity test is determined to be 0.7, which produces a distribution close to normal with the location values of the distribution converging and retains as many observations as possible. However, certain issues remain that lead to a spread in reflectivity caused by the variation in cloud properties and angle dependence, including the bidirectional reflectivity distribution of DCCs. The long-term variability in DCC reflectivity based on the updated detection method needs to be analyzed, with the results used to minimize such variation and to demonstrate the applicability of the new approach for hyperspectral UV–VIS sensors. Additionally, since the updated DCC detection can still be dependent on the calibration accuracy of the meteorological sensor such as AHI (further AMI), it must also be investigated hereafter to properly perform the DCC calibration for the environmental sensors.

**Author Contributions:** M.-H.A. designed and supervised the study; Y.L. performed the experiments, analyzed the data, and prepared the manuscript; M.K. contributed to the analysis of results. All authors contributed to the edition of the manuscript. All authors have read and agreed to the published version of the manuscript.

**Funding:** This research is supported by the Basic Science Research Program through the National Research Foundation of Korea (NRF), funded by the Ministry of Education (2018R1A6A1A08025520). Also, this research was supported by the Korea Ministry of Environment (MOE) as "Public Technology Program based Environmental Policy (2017000160002).

**Acknowledgments:** We would like to thank the National Meteorological Satellite Center (NMSC) of the Korea Meteorological Administration (KMA) for providing the AHI Level 1B data. The valuable comments from the anonymous reviewers are also greatly helpful to improve the manuscript.

**Conflicts of Interest:** The authors declare no conflict of interest.

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
