# Peer review of "The New Potential of Deep Convective Clouds as a Calibration Target for a Geostationary UV/VIS Hyperspectral Spectrometer"

_remotesensing, doi:10.3390/rs12030446_

Round 1

Reviewer 1 Report

General comments:

It is recommended to clarify and add the explanation as followings before publication:

As the satellite performance are increased, more accurate calibration is essential for achieving the final goal of the mission. And thus the results look promising and could be applied to various VIS-NIR sensors calibration.

I understand that the purpose of the manuscript is “to introduce the updated detection method of DCCs”, and proposed method could be used not only for GEMS calibration but also for meteorological satellite. But it will be better for authors to provide some preliminary results of GEMS (i.e. using proxy data) calibration, i.e. how to use the updated detection method. For example, GSICS (Global Space-based Inter-Calibration System) used the PDF of DCCs, thus both mean and mode of the DCC pixels (but usually mode of DCC pixels) would be used for monitoring the long-term trending of specific channel calibration. The proposed method has also shown the same value if we could only check the mode of DCCs in Table 4 and Table 5 in this study.

Secondly, the new method uses the AHI (or AMI future or so) for detecting DCCs. In this case the criterion (0.70, 0.018) could also be dependent (of course not much) on the calibration accuracy of AHI (and/or AMI). Why not use the GEMS data itself for detecting/selecting DCCs for calibration purpose? And then inter-calibrate with other accurate sensors.

Minor comments:

The numbers indicate the line number through the manuscript.

Please clarify the procedure of the Figures through the paper, for example the DCCs in Figures are presented “mean”, “median”, or “mode”? 124-126: The sentence is a little bit confused to understand what are “these sensors”. GEMS+TROPOMI? OMI+TROPOMI? 167: in case of freq is 0, please make the color “white” since the Figure 1 showed white color 170 and 171: it is a little bit confusion “the number of DCCs” vs. “the number of DCC pixels”. In Figure 1, the frequency is from 0 to 20 (max), while in Figure 2, the number more than 200. Please make the sentence clear. In Figure 1, clearly mentioned OMI, but please make clear OMI or TROPOMI used in Figure 2 178-180: What about including the position of 2A and 2B is the same as 128.2 while AHI is 140.7E. In Figure 3: not clear mean/SD of what area, for example, 3x3 mean, 5x5 mean and so on. 213-215: The sentence is a little confused, since Fλ is not affected by atmosphere but Iλ(θ0,θ,φ). 248: figure 5 -> Figure 5 251-252: “Thus, these pixels near the nadir port are also eliminated in this study”

please include the method and criterion how to eliminate the pixels near the nadir port.

Figure 5(b): before eliminating the pixels near the nadir port, what about after eliminating the pixels near the nadir port? I think the reader is also interested. 296: “systematic difference” means STD in Figure 6? 315: typo? DCDs -> DCCs 337: it is better to include the reason why specially use the 354nm 394: typo? Speediness -> spreadness? 467: typo? Only 500: the full name of AMI was introduced in 178-179, thus not need to repeat 501: GK-2A is not used except this sentence, please remove the abbreviation.

Author Response

Please find attached for the author response..

Best regards.

Reviewer 2 Report

This paper examines the potential of the deep convective cloud as the vicarious calibration target for GEMS in the UV-VIS spectral region. I recommend this paper to be published after minor revision.

In section 3.1, the authors use same thresholds in DCC detection for OMI and to TROPOMI. However, as stated by the authors in section 3.3.1, “Because the lower spatial resolution of OMI with the small scale DCC increases not only brightness temperature but also the spatial variability in the IR and VIS making such a pixel as a non-DCC cloud.” So why not use more relaxed thresholds for OMI, so that OMI can also detect such small scale DCC as TROPOMI? “the mean reflectivity of OMI and TROPOMI are almost identical with the higher reflectivity of about 0.9 over the whole wavelength.” In Figure 6, it seems that the mean reflectivity of TROPOMI are smaller than 0.9 over most of wavelength, and smaller than or around 0.85 for OMI. The authors compare DCC spectra of OMI and TROPOMI with quite different samples. How do the results look like if further collocate OMI and TROPOMI DCC pixels and compare with the same DCC samples? There are some differences between mean DCC spectra of OMI and TROPOMI. Will the differences be smaller if using improved DCC detection method proposed in section 4?

Author Response

Please refer attached for author response..

Best regards.
